# EFFICIENT LINK PREDICTION VIA GNN LAYERS INDUCED BY NEGATIVE SAMPLING

## ABSTRACT

Graph neural networks (GNNs) for link prediction can loosely be divided into two broad categories. First, *node-wise* architectures pre-compute individual embeddings for each node that are later combined by a simple decoder to make predictions. While extremely efficient at inference time (since node embeddings are only computed once and repeatedly reused), model expressiveness is limited such that isomorphic nodes contributing to candidate edges may not be distinguishable, compromising accuracy. In contrast, *edge-wise* methods rely on the formation of edge-specific subgraph embeddings to enrich the representation of pair-wise relationships, disambiguating isomorphic nodes to improve accuracy, but with the cost of increased model complexity. To better navigate this trade-off, we propose a novel GNN architecture whereby the *forward pass* explicitly depends on *both* positive (as is typical) and negative (unique to our approach) edges to inform more flexible, yet still cheap node-wise embeddings. This is achieved by recasting the embeddings themselves as minimizers of a forward-pass-specific energy function (distinct from the actual training loss) that favors separation of positive and negative samples. As demonstrated by extensive empirical evaluations, the resulting architecture retains the inference speed of node-wise models, while producing competitive accuracy with edge-wise alternatives.

## 1 INTRODUCTION

Link Prediction is a fundamental graph learning challenge that involves determining whether or not there should exist an edge connecting two nodes. Given the prevalence of graph-structured data, this task has widespread practical significance spanning domains such as social networks, knowledge graphs, and e-commerce, and recommendation systems (Koren et al., 2009; Chamberlain et al., 2020; Schlichtkrull et al., 2017). As one representative example of the latter, the goal could be to predict whether a user node should be linked with an item node in a product graph, where edges are indicative of some form of user/item engagement, e.g., clicks, purchases, etc.

Beyond heuristic techniques such as Common Neighbors (CN) (Barabási & Albert, 1999), Adamic-Adar (AA) (Adamic & Adar, 2003), and Resource Allocation (RA) (Zhou et al., 2009), graph neural networks (GNNs) have recently shown tremendous promise in addressing link prediction with trainable deep architectures (Kipf & Welling, 2017; Hamilton et al., 2017; Zhang & Chen, 2018; Zhang et al., 2022a; Zhu et al., 2021b; Chamberlain et al., 2023). Broadly speaking these GNN models fall into two categories, based on whether they rely on *node-wise* or *edge-wise* embeddings. The former involves using a GNN to pre-compute individual embeddings for each node that are later combined by a simple decoder to predict the presence of edges. This strategy is preferable when inference speed is paramount (as is often the case in real-world applications requiring low-latency predictions), since once node-wise embeddings are available, combining them to make predictions is cheap. Moreover, accelerated decoding techniques such as Maximum Inner Product Search (MIPS) (Shrivastava & Li, 2014; Neyshabur & Srebro, 2015; Yu et al., 2017) or Flashlight (Wang et al., 2022) exist to further economize inference. The downside though of node-wise embedding methods is that they may fail to disambiguate isomorphic nodes that combine to form a candidate edge (Zhang & Chen, 2018).

To this end, edge-wise embeddings with greater expressive power have been proposed for more robust link prediction (Zhang et al., 2022a; Yun et al., 2021; Chamberlain et al., 2023; Yin et al., 2023). These models base their predictions on edge-specific subgraphs capable of breaking isomorphic node relationships via structural information (e.g., overlapping neighbors, shortest paths, positional

encodings, or subgraph sketches) that might otherwise undermine the performance of node-wise embeddings. This flexibility comes with a substantial cost though, as inference complexity can be orders of magnitude larger given that a unique subgraph must be extracted and processed by the GNN for every test edge. Although this expense can be alleviated to some cases by pre-processing (Chamberlain et al., 2023), for inference over very large sets of candidate links, even the pre-processing time can be overwhelming relative to that required by node-wise predictors.

In this work, we address the trade-off between expressiveness and inference efficiency via the following strategy. To maintain minimal inference speeds, we restrict ourselves to a node-wise embedding approach and then try to increase the expressiveness on multiple fronts. Most importantly, we allow each node-level embedding computed during the forward pass to depend on not only its ego network (i.e., subgraph containing a target node), but also on the embeddings of negatively sampled nodes, meaning nodes that were not originally sharing an edge with the target node. This can be viewed as forming a complementary *negative* ego network for each node. Moreover, rather than heuristically incorporating the resulting positive *and* negative ego networks within a traditional GNN-based embedding model, we instead combine them so as to infuse their integration with an inductive bias specifically tailored for link prediction. Specifically, we introduce a parameterized graph-regularized energy, in the spirit of triplet ranking loss functions used for capturing both relative similarities and differences between pair-wise items. By design, the parameter-dependent minimizer of this function can then serve the role of end-to-end trainable node-wise embeddings, *explicitly dependent on the node features of both positive and negative samples even during the forward pass* (not just the backward training pass as is typical). For these reasons, we refer to our model as a *Yin* (negative) *Yang* (positive) GNN, or YinYanGNN for short.

In this way, we increase the flexibility of node-wise embedding approaches, without significantly increasing the computational complexity, as no edge-wise embeddings or edge-specific subgraph extraction is necessary. Additionally, by unifying the positive and negative samples within a single energy function minimization process, the implicit receptive field of the embeddings can be arbitrarily large without oversmoothing, a property we inherit from prior related work on optimization-based GNN models applied to much simpler node classification tasks (Yang et al., 2021). These observations lead to a statement of our primary contributions:

1. We design node-wise embeddings for link prediction that are explicitly imbued with an inductive bias informed by the node features of *both* positive and negative samples during the model forward pass. This is accomplished by recasting the embeddings themselves as minimizers of an energy function that explicitly balances the impact of positive (Yang) and negative (Yin) samples, leading to a model we refer to as YinYanGNN.

2. We analyze the convergence properties and computational complexity of the optimization process which produces YinYanGNN embeddings, as well as their expressiveness relative to traditional node-wise models. These results suggest that our approach can potentially serve as a reliable compromise between node- and edge-wise alternatives.

3. Experiments on real-world link prediction benchmarks reveal the YinYanGNN can outperform SOTA node-wise models in terms of accuracy while matching their efficiency. And analogously, YinYanGNN can exceed the efficiency of edge-wise approaches while maintaining similar (and in some cases better) prediction accuracy.

## 2 RELATED WORK

**GNNs for Link Prediction.** As mentioned in Section 1, GNN models for link prediction can be roughly divided into two categories, those based on node-wise embeddings (Kipf & Welling, 2017; Hamilton et al., 2017; Veličković et al., 2017) and those based on edge-wise embeddings (Zhang & Chen, 2018; Zhang et al., 2022a; Chamberlain et al., 2023; Yun et al., 2021; Zhu et al., 2021b; Kong et al., 2022; Yin et al., 2022; 2023). The former is generally far more efficient at inference time given that the embeddings need only be computed once for each node and then repeatedly combined to make predictions for each candidate edge. However, the latter is more expressive by facilitating edge-specific structural features at the cost of much slower inference.

**GNN Layers formed from unfolded optimization steps.** A plethora of recent research has showcased the potential of constructing resilient GNN architectures for node classification using graph

propagation layers that emulate the iterative descent steps of a graph-regularized energy function (Chen & Eldar, 2021; Liu et al., 2021; Ma et al., 2020; Pan et al., 2021; Yang et al., 2021; Zhang et al., 2020; Zhu et al., 2021a; Ahn et al., 2022). These approaches allow the node embeddings at each layer to be regarded as progressively refined approximations of an interpretable energy minimizer. A key advantage is that embeddings obtained in this way can be purposefully designed to address challenges such as GNN oversmoothing or the introduction of robustness against spurious edges. Moreover, these adaptable embeddings can be seamlessly integrated into a bilevel optimization framework (Wang et al., 2016) for supervised training. Even so, prior work in this domain thus far has been primarily limited to much simpler node classification tasks, where nuanced relationships between pairs of nodes need not be explicitly accounted for. In contrast, we are particularly interested in the latter, and the potential to design new energy functions that introduce inductive biases suitable for link prediction.

## 3 PRELIMINARIES

In this section we briefly introduce notation before providing concrete details of the link prediction problem that will be useful later.

### 3.1 NOTATION

Let $\mathcal{G} = (\mathcal{V}, \mathcal{E}, X)$ be a graph with node set $\mathcal{V}$, corresponding $d_x$-dimensional node features $X \in \mathbb{R}^{n \times d_x}$, and edge set $\mathcal{E}$, where $|\mathcal{V}| = n$. We use $A$ to denote the adjacency matrix and $D$ for the degree matrix. The associated Laplacian matrix is defined by $L \triangleq D - A$. Furthermore, $Y \in \mathbb{R}^{n \times d}$ refers to node embeddings of size $d$ we seek to learn via a node-wise link prediction procedure. Specifically the node embedding for node $i$ is $y_i$, which is equivalent to the $i$-th row of $Y$.

### 3.2 LINK PREDICTION

We begin by introducing a commonly-used loss for link prediction, which is defined over the training set $\mathcal{E}_{train} \subset \mathcal{E}$. For both node-wise and edge-wise methods, the shared goal is to obtain an edge probability score $p(v_i, v_j) = \sigma(e_{ij})$ for all edges $(v_i, v_j) \in \mathcal{E}_{train}$ (as well as negatively-sampled counterparts to be determined shortly), where $\sigma$ is a sigmoid function and $e_{ij}$ is a discriminative representation for edge $(v_i, v_j)$. Proceeding further, for every true positive edge $(v_i, v_j)$ in the training set, $N \geq 1$ negative edges $(v_{i^a}, v_{j^a})_{a=1,\dots,N}$ are randomly sampled from the graph for supervision purposes. We are then positioned to express the overall link prediction loss as

$$\mathcal{L}_{link} \triangleq \sum_{(i,j) \in \mathcal{E}_{train}} \left[ -\log(p(v_i, v_j)) - \sum_{a=1}^{N} \frac{1}{N} \log(1 - p(v_{i^a}, v_{j^a})) \right], \tag{1}$$

where each edge probability is computed with the corresponding edge representation. The lingering difference between node- and edge-wise methods then lies in how each edge representation $e_{ij}$ is actually computed.

For node-wise methods, $e_{ij} = h(y_i, y_j)$, where $y_i$ and $y_j$ are node-wise embeddings and $h$ is a decoder function ranging in complexity from a parameter-free inner-product to a multi-layer MLP. While decoder structure varies (Wang et al., 2021; Rendle et al., 2020; Sun & Wu, 2020; Hu et al., 2020; Wang et al., 2022), of particular note for its practical effectiveness is the HadamardMLP approach, which amounts to simply computing the hadamard product between $y_i$ and $y_i$ and then passing the result through an MLP. Fast, sublinear inference times are possible with the HadamardMLP using an algorithm from (Wang et al., 2022). In contrast, the constituent node embeddings themselves are typically computed with some form of trainable GNN encoder model $g$ of the form $y_i = g(x_i, \mathcal{G}_i)$ and $y_j = g(x_j, \mathcal{G}_j)$, where $\mathcal{G}_i$ and $\mathcal{G}_j$ are the subgraphs containing nodes $v_i$ and $v_j$, respectively.

Turning to edge-wise methods, the edge representation $e_{ij}$ relies on the subgraph $\mathcal{G}_{ij}$ defined by *both* $v_i$ and $v_j$. In this case $e_{ij} = h_e(v_i, v_j, \mathcal{G}_{ij})$, where $h_e$ is an edge encoder GNN whose predictions can generally *not* be decomposed into a function of individual node embeddings as before. Note also that while the embeddings from node-wise subgraphs for *all* nodes in the graph can be produced by a *single* GNN forward pass, a unique/separate edge-wise subgraph and corresponding forward pass are needed to make predictions for each candidate edge. This explains why edge-wise models endure far slower inference speeds in practice.

## 4 INCORPORATING NEGATIVE SAMPLING INTO NODE-WISE MODEL DESIGN

Previously we described how computationally-efficient node-wise embedding methods for link prediction rely on edge representations that decompose as $e_{ij} = h[g(y_i, \mathcal{G}_i), g(y_j, \mathcal{G}_j)]$ for node-pair $(v_i, v_j)$, a decomposition that is decidedly less expressive than the more general form $e_{ij} = h_e(v_i, v_j, \mathcal{G}_{ij})$ adopted by edge-wise embedding methods. Although we can never match the flexibility of the edge-wise models with a node-wise approach, we can nonetheless increase the expressiveness of node-wise models while still retaining their attractive computational footprint.

At a high-level, our strategy for accomplishing this goal is to learn node-wise embeddings of the revised form $y_i = g(v_i, \mathcal{G}_i, \mathcal{G}_i^-)$, where $\mathcal{G}_i^-$ is a subgraph of $\mathcal{G}^-$ centered at node $v_i$, $\mathcal{G}^- = (\mathcal{V}, \mathcal{E}^-, X)$, and $\mathcal{E}^-$ is a set of negatively-sampled edges between nodes in the original graph $\mathcal{G}$. In this way each node-wise embedding has access to node features from both positive and negative neighboring nodes.

To operationalize this conceptual design, rather than heuristically embedding negative samples within an existing GNN architecture (see Appendix D.3 for experiments using this simple strategy), we instead chose node-wise embeddings that minimize an energy function regularized by both positive and negative edges, i.e., $\mathcal{E}$ and $\mathcal{E}^-$. More formally, we seek a node embedding matrix $Y = \arg\min_Y \ell_{node}(\mathcal{G}, \mathcal{G}^-)$ in such a way that the optimal solution decomposes as $y_i = g(v_i, \mathcal{G}_i, \mathcal{G}_i^-)$ for some differentiable function $g$ across all nodes $v_i$. This allows us to anchor the influence of positive and negative edges within a unified energy surface, with trainable minimizers that can be embedded within the link prediction loss from (1). In the remainder of this section we motivate our selection for $\ell_{node}$, as well as the optimization steps which form the structure of the corresponding function $g$.

### 4.1 AN INITIAL ENERGY FUNCTION

Prior work on optimization-based node embeddings (Chen et al., 2021; Ma et al., 2020; Pan et al., 2021; Yang et al., 2021; Zhang et al., 2020; Zhu et al., 2021a) largely draw on energy functions related to (Zhou et al., 2004), which facilitates the balancing of local consistency relative to labels or a base predictor, with global constraints from graph structure. However, these desiderata alone are inadequate for the link prediction task, where we would also like to drive individual nodes towards regions of the embedding space where they are maximally discriminative with respect to their contributions to positive and negative edges. To this end we take additional inspiration from triplet ranking loss functions (Rendle et al., 2012) that are explicitly designed for learning representations that can capture relative similarities or differences between items.

With these considerations in mind, we initially posit the energy function

$$\ell_{node} = ||Y - f(X; W)||_F^2 + \lambda \sum_{(i,j) \in \mathcal{E}} \left[ d(y_i, y_j) - \frac{\lambda_K}{\lambda K} \sum_{j' \in \mathcal{V}_{(i,j)}^K} d(y_i, y_j') \right], \qquad (2)$$

where $f(X; W)$ (assumed to apply row-wise to each individual node feature $x_i$) represents a base model that processes the input features using trainable weights $W$, $d(y_i, y_j)$ is a distance metric, while $\lambda$ and $\lambda_K$ are hyperparameters that control the impact of positive and negative edges. Moreover, $\mathcal{V}_{(i,j)}^K$ is the set of negative destination nodes sampled for edge $(v_i, v_j)$ and $|\mathcal{V}_{(i,j)}^K| = K$. Overall, the first term pushes the embeddings towards the processed input features, while the second and third terms apply penalties to positive and negative edges in a way that is loosely related to the aforementioned triplet ranking loss (more on this below).

If we choose $d(i, j) = ||y_i - y_j||^2$ and define edges of the negative graph $\mathcal{G}^-$ as $\mathcal{E}^- \triangleq \{(i, j')| j' \in \mathcal{V}_{(i,j)}^K$ for $(i, j) \in \mathcal{E}\}$, we can rewrite (2) as

$$\ell_{node} = ||Y - f(X; W)||_F^2 + \lambda \text{tr}[Y^\top L Y] - \frac{\lambda_K}{K} \text{tr}[Y^\top L^- Y], \qquad (3)$$

where $L^-$ is the Laplacian matrix of $\mathcal{G}^-$. To find the minimum, we compute the gradients

$$\frac{\partial \ell_{node}}{\partial Y} = 2(Y - f(X; W)) + 2\lambda L Y - \frac{\lambda_K}{K} 2 L^- Y, \qquad (4)$$

where $Y^{(0)} = f(X; W)$. The corresponding gradient descent updates then become

$$Y^{(t+1)} = Y^{(t)} - \alpha((Y^{(t)} - f(X; W)) + \lambda L Y^{(t)} - \frac{\lambda_K}{K} L^- Y^{(t)}), \qquad (5)$$

where the step size is $\frac{\alpha}{2}$. We note however that (3) need not generally be convex or even lower bounded. Moreover, the gradients may be poorly conditioned for fast convergence depending on the Laplacian matrices involved. Hence we consider several refinements next to stabilize the learning process.

## 4.2 Energy Function Refinements

**Lower-Bounding the Negative Graph.** Since the regularization of negative edges brings the possibility of an ill-posed loss surface (a non-convex loss surface that may be unbounded from below) , we introduce a convenient graph-aware lower-bound analogous to the max operator used by the triplet loss. Specifically, we update (3) to the form

$$\ell_{node} = ||Y - f(X; W)||_F^2 + \lambda \text{tr}[Y^\top L Y] + \frac{\lambda_K}{K} \text{Softplus}(|\mathcal{E}|\gamma - \text{tr}[Y^\top L^- Y]), \qquad (6)$$

noting that we use $\text{Softplus}(x) = \log(1 + e^x)$ instead of $max(\cdot, 0)$ to make the energy differentiable. Unlike triplet loss that includes positive term in the $max(\cdot, 0)$ function, we only restrict the lower bound for the negative term, because we still want the positive part to impact our model when the negative part hits the bound.

**Normalization.** We use the normalized Laplacian matrices of original graph $\mathcal{G}$ and negative graph $\mathcal{G}^-$ instead to make the gradients smaller. Also, for the gradients of the first term in (3), we apply a re-scaling by summation of degrees of both graphs. The modified gradients are

$$\frac{\partial \ell_{node}}{\partial Y} = 2(D + D^-)^{-1}(Y - f(X; W)) + 2\lambda \tilde{L} Y - \frac{\lambda_K}{K} 2\tilde{L}^- Y, \qquad (7)$$

where $D$ and $D^-$ are diagonal degree matrices of $\mathcal{G}$ and $\mathcal{G}^-$. The normalized Laplacians are $\tilde{L} = D^{-\frac{1}{2}} L D^{-\frac{1}{2}}, \tilde{L}^- = D^{-\frac{1}{2}} L^- D^{-\frac{1}{2}}$ leading to the corresponding energy function

$$\ell_{node} = \left\| (D + D^-)^{-1}(Y - f(X; W)) \right\|_F^2 + \lambda \text{tr}[Y^\top \tilde{L} Y] - \frac{\lambda_K}{K} \text{tr}[Y^\top \tilde{L}^- Y]. \qquad (8)$$

**Learning to Combine Negative Graphs.** We now consider a more flexible implementation of negative graphs. More concretely, we sample $K$ negative graphs $\{\mathcal{G}_{(k)}^-\}_{k=1,\dots,K}$, in which every negative graph consists of one negative edge per positive edge $(\mathcal{G}_{(k)}^- \triangleq \{(i, j')| j' \in \mathcal{V}_{(i,j)}^1 \text{ for } (i, j) \in \mathcal{E}\})$. And we set learnable weights $\lambda_K^k$ for the structure term of each negative graph, which converts the energy function to

$$\ell_{node} = ||Y - f(X; W)||_F^2 + \lambda \text{tr}[Y^\top L Y] - \frac{1}{K} \sum_{k=1}^{K} \lambda_K^k \text{tr}[Y^\top L_k^- Y], \qquad (9)$$

Also in practice we normalize this energy function to

$$\ell_{node} = \left\| (D + D_K^-)^{-1}(Y - f(X; W)) \right\|_F^2 + \lambda \text{tr}[Y^\top \tilde{L} Y] - \frac{1}{K} \sum_{k=1}^{K} \lambda_K^k \text{tr}[Y^\top \tilde{L_k^-} Y], \qquad (10)$$

where $D_K^- = \sum_{k=1}^{K} D_k^-$, $D_k^-$ is the degree matrix of $L_k^-$ and $\tilde{L_k^-} = D_K^{-\frac{1}{2}} L_k^- D_K^{-\frac{1}{2}}$. The lower bound is also added as before. Overall, the motivation here is to inject trainable flexibility into the negative sample graph, which is useful for increasing model expressiveness.

## 4.3 The Overall Algorithm

Combining the modifications we discussed in last section (and assuming a single, fixed $\lambda_K$ here for simplicity; the more general, learnable case with multiple $\lambda_K^k$ from Section 4.2 naturally follows), we obtain the final energy function

$$\ell_{node} = \left\| (D + D^-)^{-1}(Y - f(X; W)) \right\|_F^2 + \lambda \text{tr}[Y^\top \tilde{L} Y] + \frac{\lambda_K}{K} \text{Softplus}(\gamma|\mathcal{E}| - \text{tr}[Y^\top \tilde{L}^- Y]), \qquad (11)$$

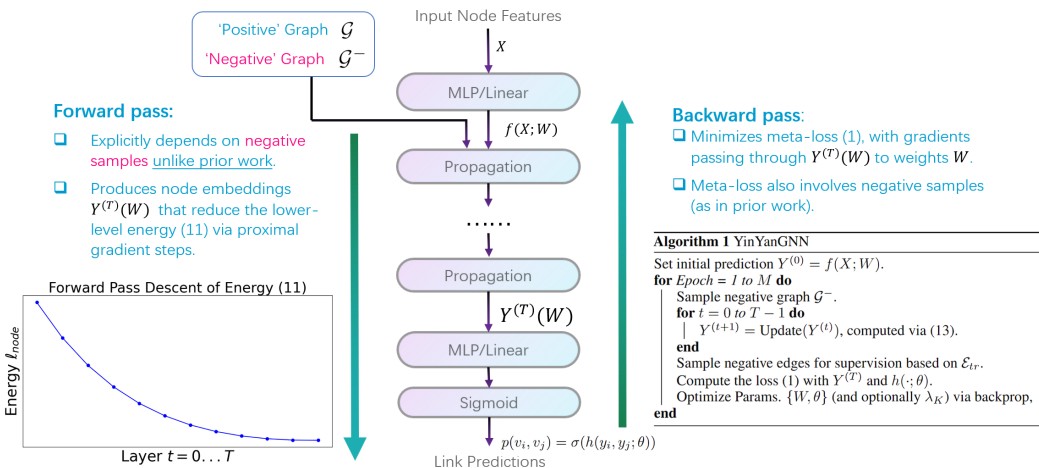

Figure 1: *YinYanGNN model illustration.* On the left side we show the YinYanGNN forward pass explicitly depending on negative samples, with layers computing embeddings that descend the lower-level energy (11). On the right side we show the more traditional backward pass for optimizing (1) and YinYanGNN pseudo-code for training; see Appendix A for further training details.

with the associated gradients

$$\frac{\partial \ell_{node}}{\partial Y} = 2(D + D^-)^{-1}(Y - f(X; W)) + 2\lambda \tilde{L} Y - \frac{\lambda_K}{K} 2\tilde{L}^- Y \sigma(Q), \qquad (12)$$

where $Q = \gamma|\mathcal{E}| - \text{tr}[Y^\top \tilde{L}^- Y]$ and $\sigma(x)$ is the sigmoid function. The final updates for our model then become

$$Y^{(t+1)} = Y^{(t)} - \alpha\Big((D + D^-)^{-1}(Y^{(t)} - f(X; W)) + \lambda \tilde{L} Y^{(t)} - \frac{\lambda_K}{K} \tilde{L}^- Y^{(t)} \sigma(Q^{(t)})\Big)$$

$$= C_1 Y^{(t)} + C_2 f(X; W) + c_3 \tilde{A} Y^{(t)} - c_4 \tilde{A}^- Y^{(t)}, \qquad (13)$$

where the diagonal scaling matrices $(C_1, C_2)$ and scalar coefficients $(c_3, c_4)$ are given by

$$C_1 = \Big(1 - \alpha\lambda + \tfrac{1}{K}\alpha\lambda_K \sigma(Q^{(t)})\Big) I - \alpha(D + D^-)^{-1}, \quad C_2 = \alpha(D + D^-)^{-1},$$

$$c_3 = \alpha\lambda, \quad c_4 = \Big(\tfrac{1}{K}\alpha\lambda_K \sigma(Q^{(t)})\Big), \qquad (14)$$

with $Q^{(t)} = \gamma|\mathcal{E}| - \text{tr}[(Y^{(t)})^\top \tilde{L}^- Y^{(t)}]$, $I$ as an $n \times n$ identity matrix, and $\frac{\alpha}{2}$ as the step size.

From the above expressions, we observe that the first and second terms of (13) can be viewed as rescaled skip connections from the previous layer and input layer/base model, respectively. As we will later show, these scale factors are designed to facilitate guaranteed descent of the objective from (11). Meanwhile, the third term of (13) represents a typical GNN graph propagation layer while the fourth term is the analogous negative sampling propagation unique to our model. In the context of Chinese philosophy, the latter can be viewed as the Yin to the Yang which is the third term, and with a trade-off parameter that can be learned when training the higher-level objective from (1), the Yin/Yang balance can in a loose sense be estimated from the data; hence the name *YinYanGNN* for our proposed approach. We illustrate key aspects of the YinYanGNN framework in Figure 1.

## 5 ANALYSIS

In this section we first address the computational complexity and convergence issues of our model before turning to further insights into the role of negative sampling in our proposed energy function.

### 5.1 TIME COMPLEXITY

YinYanGNN has a time complexity given by $O(T|\mathcal{E}|Kd + nPd^2)$ for one forward pass, where as before $n$ is the number of nodes, $T$ is the number of propagation layers/iterations, $P$ is the number of

MLP layers in the base model $f(X; W)$, and $d$ is the hidden embedding size. Notably, this complexity is of the same order as a vanilla GCN model, one of the most common GNN architectures (Kipf & Welling, 2017), which has a forward-pass complexity of $O(T(|\mathcal{E}|d + nd^2))$.

Table 1: Time complexity comparisons. For BUDDY, $b$ is the hop number for propagation and $h$ is the complexity of hash operations. $^*$ indicates that the complexity can be sublinear to $n$ via (Wang et al., 2022), an option that is only available to node-wise embedding models such as YinYanGNN.

|  | SEAL | BUDDY | YinYanGNN | GCN |
|---|---|---|---|---|
| Preprocess | $O(1)$ | $O(b|\mathcal{E}|(d+h))$ | $O(1)$ | $O(1)$ |
| Train | $O(|\mathcal{E}|d^2|\mathcal{E}_{tr}|)$ | $O((b^2h + bd^2)|\mathcal{E}_{tr}|)$ | $O(T|\mathcal{E}|Kd + nPd^2 + |\mathcal{E}_{tr}|d^2)$ | $O(T(|\mathcal{E}|d + nd^2) + |\mathcal{E}_{tr}|d^2)$ |
| Encode Decode | $O(|\mathcal{E}|d^2|\mathcal{V}_{src}|n)$ | $O((b^2h + bd^2)|\mathcal{V}_{src}|n)$ | $O(T|\mathcal{E}|Kd + nPd^2)$ $O(d^2|\mathcal{V}_{src}|n)^*$ | $O(T(|\mathcal{E}|d + nd^2))$ $O(d^2|\mathcal{V}_{src}|n)^*$ |

We now drill down further into the details of overall inference speed. We denote the set of source nodes for test-time link prediction as $\mathcal{V}_{src}$, and for each node we examine all the other nodes in the graph, which means we have to calculate roughly $|\mathcal{V}_{src}|n$ edges. We compare YinYanGNN's time with SEAL (Zhang & Chen, 2018) and a recently proposed fast baseline BUDDY (Chamberlain et al., 2023) in Table 1. Like other node-wise methods, we split the inference time of our model into two parts: computing embeddings and decoding (for SEAL and BUDDY they are implicitly combined). The node embedding computation can be done only once so it does not depend on $\mathcal{V}_{src}$. Our decoding process uses HadamardMLP to compute scores for each destination node (which can also be viewed as being for each edge) and get the top nodes (edges). From this it is straightforward to see that the decoding time dominates the overall computation of embeddings. So for the combined inference time, SEAL is the slowest because of the factor $|\mathcal{E}|n$ while BUDDY and our model are both linear in the graph node number $n$ independently of $|\mathcal{E}|$. However, BUDDY has a larger factor including the compution of subgraph structure $b^2h$, which will be much slower as our experiments will show. Moreover, unlike SEAL or BUDDY, we can apply Flashlight (Wang et al., 2022) to node-wise methods like YinYanGNN, an accelerated decoding method based on maximum inner product search (MIPS) that allows HadamardMLP to have sublinear decoding complexity in $n$. Related experiments are in Appendix B.

### 5.2 CONVERGENCE OF YINYANGNN LAYERS/ITERATIONS

Convergence criteria for the energies from (3) and (11) are as follows (see Appendix E for proofs):

**Proposition 5.1.** *If $\lambda_K < K \cdot d_{max}$, where $d_{max}$ is the largest eigenvalue of $L^-$, then (3) has a unique global minimum. Moreover, if the step-size parameter satisfies $\alpha < \left\| I + \lambda\tilde{L} - \frac{\lambda_K}{K}\tilde{L}^- \right\|_F^{-1}$, then the gradient descent iterations of (5) are guaranteed to converge to this minimum.*

**Proposition 5.2.** *There exists an $\alpha' > 0$ such that for any $\alpha \in (0, \alpha']$, the iterations (13) will converge to a stationary point of (11).*

### 5.3 ROLE OF NEGATIVE SAMPLING IN PROPOSED ENERGY FUNCTIONS

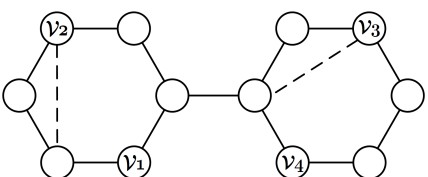

Figure 2: Modified from (Zhang et al., 2022a). Solid lines represent the original edges and dashed lines represent negative edges sampled in our model architecture (for simplicity we do not draw all negative edges).

Figure 2 serves to illustrate how the inclusion of negative samples within the forward pass of our model can potentially increase the expressiveness beyond traditional node-wise embedding approaches. As observed in the figure, $v_2$ and $v_3$ are isophormic nodes in the original graph (solid lines). However, when negative samples/edges are included the isomorphism no longer holds, meaning that link $(v_1, v_2)$ and link $(v_1, v_3)$ can be distinguished by a node-wise embedding method even without unique discriminating input features. Moreover, when combined with the flexibility of learning to balance multiple negative sampling graphs as in (10) through the trainable weights

$\{\lambda_K^k\}$, the expressiveness of YinYanGNN becomes strictly greater than a vanilla node-wise embedding method (with equivalent capacity) that has no explicit access to the potential symmetry-breaking influence of negative samples. Critically though, these negative samples are not arbitrarily inserted into our modeling framework. Rather, *they emerge by taking gradient steps (12) over a principled regularization factor (i.e., within (10) designed to push the embeddings of nodes sharing a negative edge apart during the forward pass*. In Appendix D.4 we compare this unique YinYanGNN aspect with the alternative strategy of using of random node features for breaking isomorphisms.

## 6 EXPERIMENTS

Table 2: Results on link prediction benchmarks. Baseline results are cited from prior work (Chamberlain et al., 2023; Yin et al., 2023) and the OGB leaderboard (comparisons with additional baselines can be found in the Appendix C). "-" means not reported. The format is average score $\pm$ standard deviation. The best results are bold-faced and underlined. OOM means out of GPU memory.

| | | Cora | Citeseer | Pubmed | Collab | PPA | Citation2 | DDI |
|---|---|---|---|---|---|---|---|---|
| | | HR@100 | HR@100 | HR@100 | HR@50 | HR@100 | MRR | HR@20 |
| Edge-wise GNNs | SEAL | $81.71_{\pm1.30}$ | $83.89_{\pm2.15}$ | $75.54_{\pm1.32}$ | $64.74_{\pm0.43}$ | $48.80_{\pm3.16}$ | $87.67_{\pm0.32}$ | $30.56_{\pm3.86}$ |
| | NBFnet | $71.65_{\pm2.27}$ | $74.07_{\pm1.75}$ | $58.73_{\pm1.99}$ | OOM | OOM | OOM | $4.00_{\pm0.58}$ |
| | Neo-GNN | $80.42_{\pm1.31}$ | $84.67_{\pm2.16}$ | $73.93_{\pm1.19}$ | $57.52_{\pm0.37}$ | $49.13_{\pm0.60}$ | $87.26_{\pm0.84}$ | $63.57_{\pm3.52}$ |
| | GDGNN | – | – | – | $54.74_{\pm0.48}$ | $45.92_{\pm2.14}$ | $86.96_{\pm0.28}$ | – |
| | SUREL | – | – | – | $63.34_{\pm0.52}$ | $53.23_{\pm1.03}$ | $\underline{89.74_{\pm0.18}}$ | – |
| | SUREL+ | – | – | – | $63.34_{\pm0.52}$ | $54.32_{\pm0.44}$ | $88.90_{\pm0.06}$ | – |
| | BUDDY | $88.00_{\pm0.44}$ | $92.93_{\pm0.27}$ | $74.10_{\pm0.78}$ | $65.94_{\pm0.58}$ | $49.85_{\pm0.20}$ | $87.56_{\pm0.11}$ | $78.51_{\pm1.36}$ |
| Node-wise GNNs | GCN | $66.79_{\pm1.65}$ | $67.08_{\pm2.94}$ | $53.02_{\pm1.39}$ | $44.75_{\pm1.07}$ | $18.67_{\pm1.32}$ | $84.74_{\pm0.21}$ | $37.07_{\pm5.07}$ |
| | SAGE | $55.02_{\pm4.03}$ | $57.01_{\pm3.74}$ | $39.66_{\pm0.72}$ | $48.10_{\pm0.81}$ | $16.55_{\pm2.40}$ | $82.60_{\pm0.36}$ | $53.90_{\pm4.74}$ |
| | YinYanGNN | $\mathbf{93.83_{\pm0.78}}$ | $\mathbf{94.45_{\pm0.53}}$ | $\mathbf{90.73_{\pm0.40}}$ | $\mathbf{66.10_{\pm0.20}}$ | $\mathbf{54.64_{\pm0.49}}$ | $86.21_{\pm0.09}$ | $\mathbf{80.92_{\pm3.35}}$ |

**Datasets and Evaluation Metrics.** We evaluate YinYanGNN for link prediction on Planetoid datasets: Cora (McCallum et al., 2000), Citeseer (Sen et al., 2008), Pubmed (Namata et al., 2012), and Open Graph Benchmark (OGB) link prediction datasets (Hu et al., 2020): ogbl-collab, ogbl-PPA, ogbl-Citation2 and ogbl-DDI. Planetoid represents classic citation network data, whereas OGB involves challenging, multi-domain, diverse benchmarks involving large graphs. Detailed statistics are summarized in the Appendix F. We adopt the hits ratio @k(HR@k) as the main evaluation metric as in (Chamberlain et al., 2023) for Planetoid datasets. This metric computes the ratio of positive edges ranked equal or above the $k$-th place out of candidate negative edges at test time. We set $k$ to 100 for these three datasets. For OGB datasets, we follow the official settings. Note that the metric for ogbl-Citation2 is Mean Reciprocal Rank (MRR), meaning the reciprocal rank of positive edges among all the negative edges averaged over all source nodes. Finally, we choose the test results based on the best validation results. We also randomly select 10 different seeds and report average results and standard deviation for all datasets. For details regarding hyperparameters and the implementation, please refer to Appendix F.

**Baseline Models.** To calibrate the effectiveness of our model, in the main paper we conduct comprehensive comparisons with node-wise GNNs: GCN (Kipf & Welling, 2017) and GraphSage (Hamilton et al., 2017), and edge-wise GNNs: SEAL (Zhang & Chen, 2018), NeoGNN (Yun et al., 2021), NBFnet (Zhu et al., 2021b), BUDDY (Chamberlain et al., 2023)), GDGNN (Kong et al., 2022), SUREL (Yin et al., 2022), and SUREL+ (Yin et al., 2023). We differ to Appendix C additional experiments spanning traditional link prediction heuristic methods: Common Neighbors (CN) (Barabási & Albert, 1999), Adamic-Adar (AA) (Adamic & Adar, 2003) and Resource Allocation (RA) (Zhou et al., 2009), non-GNN or graph methods: MLP, Node2vec (Grover & Leskovec, 2016), and Matrix-Factorization (MF) (Koren et al., 2009)), knowledge graph (KG) methods: TransE (Bordes et al., 2013), ComplEx (Trouillon et al., 2016), and DistMult (Yang et al., 2015), additional node-wise GNNs: GAT (Veličković et al., 2017), GIN (Xu et al., 2018a), JKNet (Xu et al., 2018b), and GC-NII (Chen et al., 2020), and finally distillation methods. Overall, for more standardized comparisons, we have chosen baselines based on published papers with open-source code and exclude those methods relying on heuristic augmentation strategies like anchor distances, or non-standard losses for optimization; such methods could be adopted by ours and others as well.

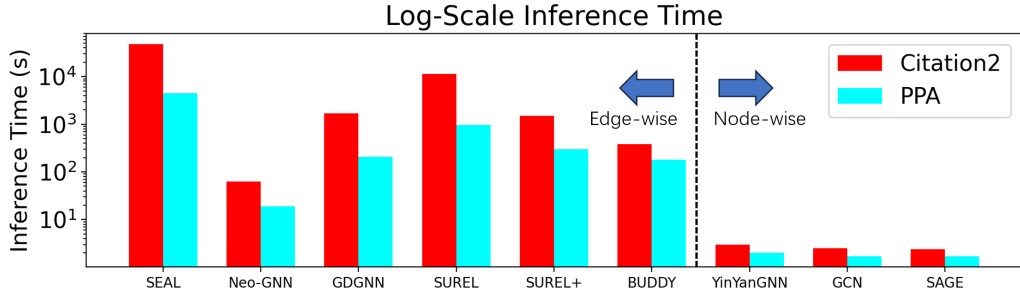

Figure 3: *Log-scale inference time.* Citation2, PPA are the two largest OGB link prediction graphs.

**Performance Results.** Accuracy results are displayed in Table 2, where we observe that YinYanGNN achieves the best performance on 6 out of 7 datasets (while remaining competitive across all 7) even when compared against more time-consuming or inference-inefficient edge-wise methods. And we outperform node-wise methods by a large margin (including several others shown in Appendix C), demonstrating that YinYanGNN can achieve outstanding predictive accuracy without sacrificing efficiency. Similarly, as included in the Appendix C, YinYanGNN also outperforms a variety of non-GNN link prediction baselines.

With regard to the latter, we next present comparisons in terms of inference speed, which is often the key factor determining whether or not a model can be deployed in real-world scenarios. For example, in an online system, providing real-time recommendations may require quickly evaluating a large number of candidate links. Figure 3 reports the results, again relative to both edge- and node-wise baseline models. Noting the log-scale time axis, from these results we observe that YinYanGNN is significantly faster than all the edge-wise models, and nearly identical to the fellow node-wise approaches as expected. And for the fastest edge-wise model, Neo-GNN, YinYanGNN is still simultaneously more efficient (Table 3) and also much more accurate (Table 2). Additionally, we remark that, as a node-wise model, the efficiency of YinYanGNN can be further improved to sublinear complexity using Flashlight (Wang et al., 2022); however, at the time of submission public Flashlight code was not available, so we differ this consideration to future work.

**Ablation over Negative Sampling.** As the integration of both positive and negative samples within a unified node-wise embedding framework is a critical component of our model, in Table 3 we report results both with and without the inclusion of the negative sampling penalty in our lower-level embedding model from (13). Clearly the former displays notably superior performance as expected. For all other ablations, we put them in Appendix D because of limited space.

Table 3: Performance of YinYanGNN with or without negative sampling in model architecture.

|  | **Cora** | **Citeseer** | **Pubmed** | **Collab** | **PPA** | **Citation2** | **DDI** |
|---|---|---|---|---|---|---|---|
| YinYanGNN | HR@100 | HR@100 | HR@100 | HR@50 | HR@100 | MRR | HR@20 |
| W/O Negative | $91.72_{\pm 0.33}$ | $92.61_{\pm 0.31}$ | $86.70_{\pm 3.23}$ | $63.02_{\pm 0.44}$ | $49.36_{\pm 2.91}$ | $83.45_{\pm 0.21}$ | $57.80_{\pm 5.39}$ |
| W/ Negative | $\mathbf{93.83_{\pm 0.78}}$ | $\mathbf{94.45_{\pm 0.53}}$ | $\mathbf{90.73_{\pm 0.40}}$ | $\mathbf{66.10_{\pm 0.20}}$ | $\mathbf{54.64_{\pm 0.49}}$ | $\mathbf{86.21_{\pm 0.09}}$ | $\mathbf{80.92_{\pm 3.35}}$ |

## 7 CONCLUSION

In conclusion, we have proposed the YinYanGNN link prediction model that achieves accuracy on par with far more expensive edge-wise models, but with the efficiency of relatively cheap node-wise alternatives. This competitive balance is accomplished using a novel node-wise architecture that incorporates informative negative samples/edges into the design of the model architecture itself to increase expressiveness, as opposed to merely using negative samples for computing a training signal as in prior work. Given the critical importance of inference speed in link prediction applications, YinYanGNN represents a promising candidate for practical usage. In terms of limitations, among other things we have not fully integrated our implementation with Flashlight for maximally accelerated inference, as public code is not yet available as previously mentioned.

## 8 REPRODUCIBILITY

We present a detailed training algorithm in Appendix A, technical proofs in Appendix E, and additional experimental/implementation details in Appendix F. Additionally, code for our model architecture is uploaded as supplemental materials with the submission; however, since ICLR is public, we choose not to release full code until after the decision.

## 9 ETHICS STATEMENT

Regarding broader societal impact, there is of course some risk that better link prediction could be used for nefarious purposes, such as recommending harmful content to minors.

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

# A  YINYANGNN ALGORITHM

We summarize the entire YinYanGNN pipeline for link prediction in Algorithm 1.

---
**Algorithm 1** YinYanGNN Bilevel Optimization Algorithm for Link Prediction

---
**Input:** Graph $\mathcal{G}$, node features $X$, number of layers $T$, number of epochs $M$, trainable base model $f(\cdot; W)$, HadamardMLP predictor $h(\cdot; \theta)$

**Output:** Trained model weights $W$ and Hadamard MLP predictor weights $\theta$.

1  Set initial prediction $Y^{(0)} = f(X; W)$, where $f$ is the trainable base model.

   **for** *Epoch = 1 to M* **do**

2      Sample negative graph $\mathcal{G}^-$.

      **for** $t = 0$ *to* $T - 1$ **do**

3         $Y^{(t+1)} = \text{Update}(Y^{(t)})$, computed via (13).

4      **end**

5      Sample negative edges for supervision based on $\mathcal{E}_{tr}$.

      Compute the link prediction loss (1) with node embeddings $Y^{(T)}$ and HadamardMLP $h(\cdot; \theta)$.

      Backpropagate over all parameters using optimizer (Adam, SGD, etc.)

6  **end**

---

# B  FURTHER INFERENCE TIME COMPARISONS

In the main paper we have compared full inference times. However, in practice, we can save the embeddings output from the encoder and only compare the decoding step, which is a key differentiator. To this end, we compare the decoding speed of YinYanGNN with SEAL, a highly-influential edge-wise model, and BUDDY, which represents a strong baseline with a good balance between accuracy and speed for edge-wise GNNs. We conduct the experiments according to (Wang et al., 2022). Table 4 displays the results, where our model achieves the fastest performance compared with these two baselines. We also note that although the time complexity of BUDDY is linear in $n$ (which is the same as our model without Flashlight), the different scaling coefficients caused by the required subgraph information calculation still make BUDDY much slower than YinYanGNN.

Table 4: Decoding speed at inference time on Citation2 as measured by nodes/sec. (higher is better).

|  | SEAL | BUDDY | YinYanGNN | YinYanGNN(dot) | YinYanGNN(dot MIPS) |
|---|---|---|---|---|---|
| nodes/second | 0.00024 | 0.005 | 0.6 | 1.3 | 25 |

Finally, although the code for Flashlight is not yet publically available, we can mimic the acceleration it will produce as follows. As Flashlight is a repeated version of MIPS, which can be applied to dot product prediction, we also run MIPS on our model with dot product as the decoder to see how fast MIPS can accelerate. We run the experiments on CPU and achieve a 20x speedup (compare last two columns of Table 4 with and without MIPS).

# C  COMPARISONS WITH ADDITIONAL BASELINE MODELS

In this section we compare YinYanGNN with more diverse classical baselines, expressive node-wise GNNs, and distillation methods.

## C.1  NON-GNN APPROACHES

Herein we show results for classical approaches reported from Chamberlain et al. (2023) and OGB leaderboards, with the exception that results for MLP and Node2vec for Cora, Citeseer and Pubmed are obtained by our own implementation in Appendix F. These new baselines include traditional heuristic methods (Common Neighbors (CN) (Barabási & Albert, 1999), Adamic-Adar

(AA) (Adamic & Adar, 2003) and Resource Allocation (RA) (Zhou et al., 2009) ), non-GNN methods (MLP, Node2vec (Grover & Leskovec, 2016), Matrix-Factorization (MF) (Koren et al., 2009)), and knowledge graph (KG) methods(TransE (Bordes et al., 2013), ComplEx (Trouillon et al., 2016), DistMult (Yang et al., 2015)). YinYanGNN performance exceeds that of these methods.

Table 5: Results on link prediction benchmarks. Baseline results are cited from previous works (Chamberlain et al., 2023; Yin et al., 2023) and the OGB leaderboard, with a few exceptions mentioned in the text.

| | | Cora | Citeseer | Pubmed | Collab | PPA | Citation2 | DDI |
|---|---|---|---|---|---|---|---|---|
| | | HR@100 | HR@100 | HR@100 | HR@50 | HR@100 | MRR | HR@20 |
| Heuristics | CN | $33.92_{\pm0.46}$ | $29.79_{\pm0.90}$ | $23.13_{\pm0.15}$ | $56.44_{\pm0.00}$ | $27.65_{\pm0.00}$ | $51.47_{\pm0.00}$ | $17.73_{\pm0.00}$ |
| | AA | $39.85_{\pm1.34}$ | $35.19_{\pm1.33}$ | $27.38_{\pm0.11}$ | $64.35_{\pm0.00}$ | $32.45_{\pm0.00}$ | $51.89_{\pm0.00}$ | $18.61_{\pm0.00}$ |
| | RA | $41.07_{\pm0.48}$ | $33.56_{\pm0.17}$ | $27.03_{\pm0.35}$ | $64.00_{\pm0.00}$ | $49.33_{\pm0.00}$ | $51.98_{\pm0.00}$ | $27.60_{\pm0.00}$ |
| Non-GNN | MLP | $16.66_{\pm0.24}$ | $21.49_{\pm0.30}$ | $24.85_{\pm1.14}$ | $19.27_{\pm1.29}$ | $0.46_{\pm0.00}$ | $29.06_{\pm0.16}$ | NA |
| | Node2vec | $46.87_{\pm0.97}$ | $50.76_{\pm2.32}$ | $54.40_{\pm11.10}$ | $48.88_{\pm0.54}$ | $22.26_{\pm0.88}$ | $61.41_{\pm0.11}$ | $23.26_{\pm2.09}$ |
| | MF | $37.37_{\pm0.00}$ | $48.96_{\pm0.02}$ | $13.79_{\pm11.97}$ | $38.86_{\pm0.29}$ | $32.29_{\pm0.94}$ | $51.86_{\pm4.43}$ | $13.68_{\pm4.75}$ |
| KG | transE | $67.40_{\pm1.60}$ | $60.19_{\pm1.15}$ | $36.67_{\pm0.99}$ | $29.40_{\pm1.15}$ | $22.69_{\pm0.49}$ | $76.44_{\pm0.18}$ | $6.65_{\pm0.20}$ |
| | complEx | $37.16_{\pm2.76}$ | $42.72_{\pm1.68}$ | $37.80_{\pm1.39}$ | $53.91_{\pm0.50}$ | $27.42_{\pm0.49}$ | $72.83_{\pm0.38}$ | $8.68_{\pm0.36}$ |
| | DistMult | $41.38_{\pm2.49}$ | $47.65_{\pm1.68}$ | $40.32_{\pm0.89}$ | $51.00_{\pm0.54}$ | $28.61_{\pm1.47}$ | $66.95_{\pm0.40}$ | $11.01_{\pm0.49}$ |
| Node-wise GNN | YinYanGNN | $\mathbf{93.83_{\pm0.78}}$ | $\mathbf{94.45_{\pm0.53}}$ | $\mathbf{90.73_{\pm0.40}}$ | $\mathbf{66.10_{\pm0.20}}$ | $\mathbf{54.64_{\pm0.49}}$ | $\mathbf{86.21_{\pm0.09}}$ | $\mathbf{80.92_{\pm3.35}}$ |

## C.2 MORE NODE-WISE GNNS

We also include experiments with more expressive GNN architectures, namely GAT (Veličković et al., 2017), GIN (Xu et al., 2018a), JKNet (Xu et al., 2018b), and GCNII (Chen et al., 2020), that all can be applied to arbitrary graph structure and have readily available public implementations. We tune these baselines according to our implementation in Appendix F. Results are shown in the table below, where we observe that YinYanGNN still maintains its strong advantage.

Table 6: Comparison of YinYanGNN with additional node-wise GNNs.

| | Cora | Citeseer | Pubmed |
|---|---|---|---|
| Model | HR@100 | HR@100 | HR@100 |
| GCNII | $48.11_{\pm1.86}$ | $44.23_{\pm1.55}$ | $49.72_{\pm2.39}$ |
| GIN | $57.23_{\pm2.87}$ | $69.19_{\pm0.41}$ | $50.97_{\pm1.61}$ |
| JK-Net | $67.86_{\pm2.32}$ | $54.04_{\pm3.73}$ | $62.73_{\pm4.53}$ |
| GAT | $79.10_{\pm0.41}$ | $64.56_{\pm2.88}$ | $41.81_{\pm2.01}$ |
| YinYanGNN | $\mathbf{93.83_{\pm0.78}}$ | $\mathbf{94.45_{\pm0.53}}$ | $\mathbf{90.73_{\pm0.40}}$ |

## C.3 DISTILLATION METHODS

Another promising method for faster inference time is to distill GNN models. Such distillation methods are complementary to our work (which mainly focuses on GNN architecture design), and could in principle be applied in parallel using YinYanGNN as a teacher model. While we reserve such an effort to future work, for now we compare performance of existing distillation pipelines including LLP (Guo et al., 2023) and two baseline distillation models, logit matching (Zhang et al., 2022b) and representation matching. Results are in Table 7, where the splits for Cora, Citeseer and Pubmed follow (Guo et al., 2023), while for Collab we use official settings. From the table we observe that YinYanGNN performance is considerably higher.

# D ABLATIONS

In this section we present some ablations of YinYanGNN.

Table 7: Results compared with distillation methods.

|  | **Cora** | **Citeseer** | **Pubmed** | **Collab** |
|---|---|---|---|---|
| **Model** | HR@20 | HR@20 | HR@20 | HR@50 |
| **Logit Matching** | $74.72_{\pm4.27}$ | $72.44_{\pm1.52}$ | $42.78_{\pm3.15}$ | $35.97_{\pm0.96}$ |
| **Representation Matching** | $75.75_{\pm1.51}$ | $65.19_{\pm5.54}$ | $44.44_{\pm2.40}$ | $36.86_{\pm0.45}$ |
| **LLP** | $78.82_{\pm1.74}$ | $77.32_{\pm2.42}$ | $57.33_{\pm2.42}$ | $49.10_{\pm0.57}$ |
| **YinYanGNN** | $\mathbf{85.39}_{\pm\mathbf{2.38}}$ | $\mathbf{84.96}_{\pm\mathbf{2.51}}$ | $\mathbf{61.02}_{\pm\mathbf{6.87}}$ | $\mathbf{66.10}_{\pm\mathbf{0.20}}$ |

## D.1 Learning Negative Graphs

One benefit of learning $\{\lambda_K^k\}$ is that it allows us to better match the performance of larger $K$ with fixed $\lambda_K$ using a smaller $K$ but with learnable $\{\lambda_K^k\}$. The latter is more practical given the greater efficiency of a smaller $K$. Figures 4 and 5 demonstrate this phenomena, whereby a learnable $\{\lambda_K^k\}$ achieves better accuracy for smaller values of $K$ on Cora and Pubmed datasets (for larger $K$ performance is similar).

For bigger datasets where using larger $K$ can be prohibitively expensive, this capability can be exploited to achieve higher accuracy. For example, on PPA and Citation2 a larger $K$ may well improve accuracy, but this is not computationally feasible without sampling or other approximations. In contrast, we can improve performance with small $K$ just by learning $\{\lambda_K^k\}$ as shown in Table 8. (In contrast, for dense graphs a larger $K$ is less likely to be helpful, in part because the extra edges will make the negative graph much more dense and more similar to complete graph, which results in the loss of structural information. Not surprisingly then, we found that learning $\{\lambda_K^k\}$ did not improve performance on the dense graph DDI dataset where $K = 1$ was optimal.)

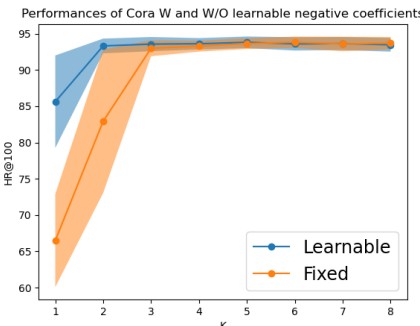

Figure 4: Learnable or Fixed $\{\lambda_K^k\}$ performances on Cora, standard deviation shown by backgrounds.

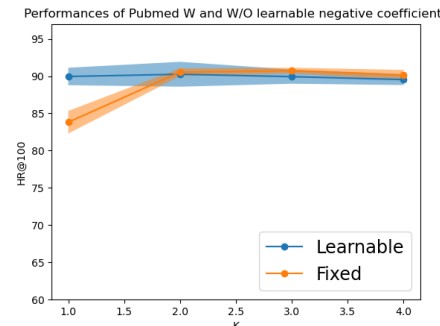

Figure 5: Learnable or Fixed $\{\lambda_K^k\}$ performances on Pubmed, standard deviation shown by backgrounds.

Table 8: Better Performances on PPA and Citation2 with learnable $\{\lambda_K^k\}$

|  | **PPA** | **Citation2** |
|---|---|---|
|  | HR@100 | MRR |
| **fixed** $\{\lambda_K^k\}$ | $52.32_{\pm0.24}$ | $85.80_{\pm0.08}$ |
| **learnable** $\{\lambda_K^k\}$ | $\mathbf{54.64}_{\pm\mathbf{0.49}}$ | $\mathbf{86.21}_{\pm\mathbf{0.09}}$ |

## D.2 Different Negative Samplers

We show the results using different negative samplers in YinYanGNN architecture. As shown in Table 9, our model is tolerant to negative samplers from sampling negative edges uniformly in the

graph (namely global uniform sampler) to sampling negative edges by fixing the source node and sampling negative destination nodes (namely source uniform sampler). .

Table 9: Performance of YinYanGNN with different samplers. Global means global uniformly sampler. Source means source uniformly sampler.

| | Cora | Citeseer | Pubmed |
|---|---|---|---|
| Sampler | HR@100 | HR@100 | HR@100 |
| Global | $92.63_{\pm 1.17}$ | $93.43_{\pm 0.49}$ | $90.54_{\pm 0.49}$ |
| Source | $92.69_{\pm 0.21}$ | $94.45_{\pm 0.53}$ | $90.40_{\pm 0.28}$ |

### D.3 NEGATIVE SAMPLING IN THE FORWARD PASS OF COMMON GNNS

As we discussed in Section 4 of the main paper, heuristically embedding negative samples within an existing GNN architecture is another possible strategy. But in this way the negative sampling terms are not integrated by a principled energy function. We follow the analysis of negative sampling in energy function, and modify the GCN architecture to incorporate negative sampling in the forward model as follows

$$Y^{(t+1)} = \text{ReLU}((\tilde{A} - \frac{\lambda_K}{K}\tilde{A}^-)Y^{(t)}W^{(t)}), \quad (15)$$

where $W^{(t)}$ are the layer-wise weights and $K$ is the negative edge number for each positive edge. Also, $\lambda_K$ is the hyperparameter to control the impact of negative edges as before. As shown in Table 10, negative sampling is also useful in some datasets for GCNs, but the performance is still not as good compared to YinYanGNN, where the negative samples are anchored to the proposed energy function.

Table 10: Performance of GCN with negative edge sampling, compared with vanilla GCN and YinYanGNN.

| | Cora | Citeseer | Pubmed |
|---|---|---|---|
| model | HR@100 | HR@100 | HR@100 |
| GCN | $66.79_{\pm 1.65}$ | $67.08_{\pm 2.94}$ | $53.02_{\pm 1.39}$ |
| GCN W/ negative | $76.15_{\pm 0.69}$ | $67.80_{\pm 3.74}$ | $74.96_{\pm 1.15}$ |
| YinYanGNN | $\mathbf{93.83_{\pm 0.78}}$ | $\mathbf{94.45_{\pm 0.53}}$ | $\mathbf{90.73_{\pm 0.40}}$ |

### D.4 RANDOM FEATURES VERSUS NEGATIVE EDGES

In Section 5.3 of the main paper, we show that random edges can in principle break the stated isomorphism. However, another potential way to accomplish this effect is to use random features that disambiguate each node. However, this can introduce undesirable auxilary effects, which become apparent when we analyze the resulting energy function when applying random features instead of negative sampling.

When we add random features to (3) we obtain the modified form

$$\ell_{node_r}(Y) = \|Y - f(X + X_R; W)\|_F^2 + \lambda \text{tr}[Y^T LY]. \quad (16)$$

For simplicity of illustration, we consider an $f$ that obeys the distributive property, i.e., $f(X + X_R; W) = f(X; W) + f(X_R; W)$. We then compute the gradient

$$\frac{\partial \ell_{node_r}}{\partial Y} = 2(Y - f(X + X_R; W)) + 2\lambda LY \quad (17)$$
$$= 2(Y - f(X; W)) + 2\lambda LY - 2f(X_R; W).$$

From these expressions we observe that random features merely introduce random blurriness to the gradient updates; they do not emerge from an otherwise useful regularization factor. In contrast, with YinYanGNN, the negative samples create gradients that push the node embeddings of nodes that do not share a true edge apart, a useful form of structural regularization that is independent of any mechanism to break isomorphisms per se.

We support these points with additional experiments in Table 11, where random features are introduced and compared against our original YinYanGNN model. From these results we observe that random features can actually degrade performance and introduce instabilities as evidenced by the reduced accuracy and larger standard deviations.

Table 11: Performance of YinYanGNN variants.

|  | Cora | Citeseer | Pubmed |
|---|---|---|---|
| YinYanGNN | HR@100 | HR@100 | HR@100 |
| W/O Negative edges | $91.72_{\pm 0.33}$ | $92.61_{\pm 0.31}$ | $86.70_{\pm 3.23}$ |
| W/ Random features | $91.25_{\pm 1.41}$ | $89.69_{\pm 0.78}$ | $79.82_{\pm 5.15}$ |
| W/ Negative edges | $\mathbf{93.83}_{\pm 0.78}$ | $\mathbf{94.45}_{\pm 0.53}$ | $\mathbf{90.73}_{\pm 0.40}$ |

# E PROOFS

## E.1 PROOF FOR PROPOSITION 5.1

We first restate Proposition 5.1 here.

**Proposition E.1.** *If $\lambda_K < K \cdot d_{max}$, where $d_{max}$ is the largest eigenvalue of $L^-$, then (3) has a unique global minimum. Moreover, if the step-size parameter satisfies $\alpha < \left\| I + \lambda\tilde{L} - \frac{\lambda_K}{K}\tilde{L}^- \right\|_F^{-1}$, then the gradient descent iterations of (5) are guaranteed to converge to this minimum.*

*Proof.* We first demonstrate conditions whereby (3) is strongly-convex, in which case it will have a unique global minimum. We can accomplish this by showing that the smallest eigenvalue of the corresponding Hessian Matrix is non-negative.[1] In our case, the smallest eigenvalue can be computed as $1 + \lambda d_{min}^+ - \frac{\lambda_K}{K}d_{max}$, where $d_{min}^+ = 0$ is the smallest eigenvalue of $L$, and $d_{max}$ is the largest eigenvalue of $L^-$, which is non-negative. So to guarantee strong convexity, we require that $1 - \frac{\lambda_K}{K}d_{max} > 0$ or equivalently that $\lambda_K < K \cdot d_{max}$.

Proceeding further, if the gradient of a strongly-convex function is Lipschitz-continuous, with Lipschitz constant $C$, it follows that gradient descent with step-size less than $C^{-1}$ will converge to the unique global minimum. In our setting, (3) has Lipschitz continuous gradients with Lipschitz constant that can be inferred via the bound

$$\left\| \frac{\partial \ell_{node}}{\partial Y_1} - \frac{\partial \ell_{node}}{\partial Y_2} \right\|_F^2 = \left\| 2 \left[ Y_1 - Y_2 + \lambda\tilde{L}(Y_1 - Y_2) - \frac{\lambda_K}{K}\tilde{L}^-(Y_1 - Y_2) \right] \right\|_F^2 \qquad (18)$$

$$\leq \left\| 2 \left[ I + \lambda\tilde{L} - \frac{\lambda_K}{K}\tilde{L}^- \right] \right\|_F^2 \|Y_1 - Y_2\|_F^2,$$

which holds for any pair of points $\{Y_1, Y_2\}$. From this it follows that $C = \left\| 2 \left[ I + \lambda\tilde{L} - \frac{\lambda_K}{K}\tilde{L}^- \right] \right\|_F$, and hence, if our assumed step-size $\frac{\alpha}{2}$ is chosen such that $\alpha < \left\| I + \lambda\tilde{L} - \frac{\lambda_K}{K}\tilde{L}^- \right\|_F^{-1}$, the result is established. $\square$

## E.2 PROOF FOR PROPOSITION 5.2

We first restate the Proposition 5.2 here before the proof.

---

[1] See for example, Dimitri Bertsekas. *Convex Optimization Algorithms*. Athena Scientific, 2015.

**Proposition E.2.** *There exists an $\alpha' > 0$ such that for any $\alpha \in (0, \alpha']$, the iterations (13) will converge to a stationary point of (11).*

*Proof.* We first show that the energy from (11) has a Lipschitz continuous gradient within the compact set $\|Y\|_F^2 \leq b^2$, where $b > 0$ is some constant and the associated (local) Lipschitz constant denoted $C_b$ will be a function of this $b$. To show this, we begin by calculating the secondary gradient for (11). For this purpose we first write the gradient in the element-wise form

$$\frac{\partial \ell_{node}}{\partial Y_{ij}} = 2(D + D^-)^{-1}(Y_{ij} - f(X;W)_{ij}) + 2\lambda \sum_k \tilde{L}_{ik} Y_{kj} - 2\frac{\lambda_K}{K} \sum_k \tilde{L}_{ik}^- Y_{kj} \sigma(Q). \quad (19)$$

So the secondary gradient is

$$\frac{\partial^2 \ell_{node}}{\partial Y_{ij}^2} = 2(D + D^-)_{ij}^{-1} Y_{ij} + 2\lambda \tilde{L}_{ii} - 2\frac{\lambda_K}{K} \tilde{L}_{ii}^- \sigma(Q) - 2\frac{\lambda_K}{K} \tilde{L}_{ii}^- Y_{ij} \sigma(Q)(1 - \sigma(Q)) \frac{\partial Q}{\partial Y_{ij}}. \quad (20)$$

Recall that $Q = \gamma|\mathcal{E}| - \text{tr}[Y^\top \tilde{L}^- Y]$, so

$$\frac{\partial Q}{\partial Y_{ij}} = -2 \sum_k \tilde{L}_{ik}^- Y_{kj}. \quad (21)$$

Consequently the secondary gradient becomes

$$\frac{\partial^2 \ell_{node}}{\partial Y_{ij}^2} = 2(D + D^-)_{ij}^{-1} Y_{ij} + 2\lambda \tilde{L}_{ii} - 2\frac{\lambda_K}{K} \tilde{L}_{ii}^- \sigma(Q) - 2\frac{\lambda_K}{K} \tilde{L}_{ii}^- \sigma(Q)(1 - \sigma(Q))(-2 \sum_k \tilde{L}_{ik}^- Y_{kj}) Y_{ij} \quad (22)$$

$$\leq 2|(D + D^-)_{ij}^{-1}|b + 2\lambda|\tilde{L}_{ii}| + |2\frac{\lambda_K}{K} \tilde{L}_{ii}^-| + 4\frac{\lambda_K}{K}|\tilde{L}_{ii}^-|(\sum_k |\tilde{L}_{ik}^-|b^2),$$

where $|\cdot|$ denotes the absolute value of scalar-valued quantities. The above inequality comes from the fact that for any $\|Y\|_F^2 \leq b^2$, $|Y_{ij}| \leq b$ and $|\sum_k \tilde{L}_{ik}^- Y_{kj} Y_{ij}| \leq \sum_k |\tilde{L}_{ik}^- Y_{kj} Y_{ij}| \leq \sum_k |\tilde{L}_{ik}^-|b^2$. Besides, $\sigma(Q) < 1$ and $1 - \sigma(Q) < 1$. Since elements of the Hessian exist and are bounded, the corresponding gradients must be Lipschitz continuous within the stated constraint set.

Next, we note that for any initialization $Y^{(0)}$ we can choose a $b$ sufficiently large such that $\ell_{node}(Y) > \ell_{node}(Y^{(0)})$ for all $Y \in \mathcal{S}_b \triangleq \{Y' : Y' \in \mathbb{R}^{n \times d}, \|Y'\|_F^2 > b^2\}$. To see this, note that for any $Y$ with $\|Y\|_F^2$ sufficiently large, $\ell_{node}(Y)$ is unbounded from above given that the first two terms collectively are strongly convex, while the last term is bounded from below.

We then define $\bar{\mathcal{S}}_b = R^{n \times d} \setminus \mathcal{S}_b$ (i.e., the complement of $\mathcal{S}_b$) for convenience. Therefore, given any initialization $Y^{(0)}$, there will exist a Lipschitz constant $C_b$ such that any gradient descent iteration computed at points $Y \in \bar{\mathcal{S}}_b$ using a step-size $\frac{\alpha}{2}$ less than $1/C_b$ (so $\alpha' \triangleq 2/C_b$) is guaranteed to reduce $\ell_{node}(Y)$, and in doing so, the iteration will remain within $\bar{\mathcal{S}}_b$. Hence we can apply standard results[2] for the convergence of gradient descent applied to non-convex functions $f$ to a stationary point provided that $f$ is lower-bounded (as is (11)) and has Lipschitz-continuous gradients, and the step-size parameter is chosen to be less than the inverse of the corresponding Lipschitz constant. □

## F  FURTHER EXPERIMENTAL DETAILS

### F.1  DATASETS STATISTICS

Statistics of the datasets used in the main paper are presented in Table 12. The splits for Cora, Citeseer and Pubmed are 7:1:2 for training:validation:test, following (Chamberlain et al., 2023). Splits for OGB datasets are the official ones from (Hu et al., 2020).

---

[2] Dimitri Bertsekas. *Nonlinear Programming*. Athena Scientific, 2nd edition, 1999.

Table 12: Datasets Statistics.

|  | Cora | Citeseer | Pubmed | Collab | PPA | DDI | Citation2 |
|---|---|---|---|---|---|---|---|
| Node number | 2,708 | 3,327 | 18,717 | 235,868 | 576,289 | 4,267 | 2,927,963 |
| Edge number | 5,278 | 4,676 | 44,327 | 1,285,465 | 30,326,273 | 1,334,889 | 30,561,187 |
| splits | rand | rand | rand | time | throughput | time | protein |
| avg $degree$ | 3.9 | 2.74 | 4.5 | 5.45 | 52.62 | 312.84 | 10.44 |

## F.2 IMPLEMENTATION AND HYPERPARAMETERS

The experiments were conducted on RTX 4090(24GB) and A100(40GB) (A100 only for ogbl-Citation2). The inference time comparison is conducted on RTX A10G. Hyperparameters were selected using a grid search on Cora, Citeseer and Pubmed. On OGB datasets, we empirically try different hyperparameters based on the experiences from the small datasets. The searching grid is the same for our model and baseline experiments if needed. For these we follow consistent ranges with the code from prior work for YinYanGNN, i.e., learning rate (0.0001-0.01), hidden dimension (32-1024), dropout (0-0.8). For specified hyperparameters for YinYanGNN, the searching grid is K (1-4), propagation step (2-16), $\alpha$ (0.01-1), $\lambda_K$ (100-1 or Learnable), $\lambda$ (100-1).

