# OpenReview forum: "Efficient Link Prediction via GNN Layers Induced by Negative Sampling"
_ICLR.cc/2024/Conference — ICLR 2024 Conference Withdrawn Submission_

### Official Review · Reviewer_32eE · 2023-10-31

**Soundness:** 3 good
**Presentation:** 2 fair
**Contribution:** 2 fair
**Rating:** 5
**Confidence:** 4

**Summary:**

The proposed method tackles the link prediction tasks by combining positive and negative edges in the forward pass, resulting in a fast and accurate model.

**Strengths:**

1. The authors provide the time complexity analysis and show the convergence of the proposed method.

2. The experimental results show that the proposed method outperforms most of the baseline methods across multiple benchmark datasets.

**Weaknesses:**

1. The major contribution of this paper is to include the negative subgraph during the node representation learning. However, my major concern is how to avoid sampling the false-negative edges during the sampling strategy. These false-negative edges will greatly influence the performance of the proposed method.

2. The presentation of this paper could be improved by breaking some very long sentences into two or more. For instance, "However, these desiderata alone are inadequate for the link prediction task, where we would also like to drive individual nodes towards regions of the embedding space where they are maximally discriminative with respect to their contributions to positive and negative edges.". In addition, the comma is missing in many sentences. For instance, "However, when negative samples/edges are included the isomorphism no longer holds."

3. Only part of the code is provided. It's impossible to reproduce the experimental results.

4. The experimental setting is not clear. See question 1 below.

5. In Table 2, the highlighted MRR score (SEAL) on the Citation2 dataset is not the best one.

**Questions:**

1. I am curious about the experimental setup. How do you determine the training set, validation set, and the test set? Do you mask a certain percentage of edges as validation and test sets in the experiment? If so, how do you avoid sampling the positive edges using negative sampling?

2. Table 3 shows the ablation study over negative sampling. I am curious why "W/O Negative" is still significantly better than most other baseline methods listed in Table 2 across multiple datasets, especially for those edge-based methods. Since the major contribution of the proposed method is to embed the negative samples into the learned representations for the link prediction, the proposed method with removed negative sampling should have a similar performance to node-wise methods.

3. Should $j'\in V^1_{(i,j)}$ be $j'\in V^k_{(i,j)}$ right above equation 9?

---

> ### Author Response · Authors · 2023-11-19
> **Response to Reviewer 32eE (part 1)**
>
> **Comment:**
> *Weakness 1: The major contribution of this paper is to include the negative subgraph during the node representation learning. However, my major concern is how to avoid sampling the false-negative edges during the sampling strategy. These false-negative edges will greatly influence the performance of the proposed method.*
>
> **Response:**
> While our *use* of negative samples is fundamentally different from prior work, the way these samples are actually obtained is identical to existing link prediction methods.  This means that we avoid sampling false-negatives the same way any other prior link prediction framework does. In practice, we use the public negative sampling code from the Deep Graph Library (DGL). This negative sampling implementation explicitly considers the original graph to avoid sampling false-negative edges. Although this incurs some time cost, because the negative sampling occurs only once per epoch, it is in irrelevant factor w.r.t. the overall training time.
>
> In any event, this issue was mentioned as the reviewer's major concern, so we hope the above provides a resolution.  If not, we are happy to provide further details as needed.
>
> **Comment:**
> *Weakness 2: The presentation of this paper could be improved by breaking some very long sentences into two or more. For instance, "However, these desiderata alone are inadequate for the link prediction task, where we would also like to drive individual nodes towards regions of the embedding space where they are maximally discriminative with respect to their contributions to positive and negative edges.". In addition, the comma is missing in many sentences. For instance, "However, when negative samples/edges are included the isomorphism no longer holds."*
>
> **Response:**
> Thanks for the suggestions, we can update the draft accordingly.
>
> **Comment:**
> *Weakness 3: Only part of the code is provided. It's impossible to reproduce the experimental results.*
>
> **Response:**
> Because ICLR submission materials are made entirely public for anyone to see and use, we did not feel comfortable providing full reproducible code while still under review.  However, we have still provided the code underlying our core model for evaluation purposes during the review process if needed.
>
> **Comment:**
> *Weakness 4: The experimental setting is not clear. See question 1 below.*
>
> **Response:**
> See our response to question 1 below.
>
>
> **Comment:**
> *Weakness 5: In Table 2, the highlighted MRR score (SEAL) on the Citation2 dataset is not the best one.*
>
> **Response:**
> Thank you for pointing out this oversight; we have corrected it.
>
> **Comment:**
> *Question 1: I am curious about the experimental setup. How do you determine the training set, validation set, and the test set? Do you mask a certain percentage of edges as validation and test sets in the experiment? If so, how do you avoid sampling the positive edges using negative sampling?*
>
> **Response:**
> For Cora, Citeseer and Pubmed, the training set, validation set, and the test set are sampled using the same splitting percentage as in (Chamberlain et al., 2023), and the validation and test edges are masked during training exactly as in prior work. For every result, we run on 10 random seeds (where each seed leads to different random set splits) and report average performances and standard deviation. For the OGB datasets, the training, validation, and test sets are fixed by the official OGB team. We also run on 10 random seeds and report average performance and standard deviation.
>
> For negative sampling, we use the public code from the Deep Graph Library (DGL). This negative sampling implementation considers the original graph to avoid sampling false-negative edges. This incurs some time cost of course; however, because the negative sampling occurs only once per epoch, it is in irrelevant factor w.r.t. the overall training time.

---

> ### Author Response · Authors · 2023-11-19
> **Response to Reviewer 32eE (part 2)**
>
> **Comment:**
> *Question 2: Table 3 shows the ablation study over negative sampling. I am curious why "W/O Negative" is still significantly better than most other baseline methods listed in Table 2 across multiple datasets, especially for those edge-based methods. Since the major contribution of the proposed method is to embed the negative samples into the learned representations for the link prediction, the proposed method with removed negative sampling should have a similar performance to node-wise methods.*
>
> **Response:**
> Actually, even without the negative sampling incorporated into the energy model, unfolded GNNs can be quite effective, and to our knowledge, they have not been extensively studied in the context of link prediction.  In some sense then, this demonstration could be viewed as a secondary/ancillary contribution of our work.  One potential reason why models designed in this way can be effective, even without negative sampling, may be that each node embedding is influenced by a large implicit receptive field, a consequence of the energy being defined over the whole graph, with no decoupling over subgraphs.
>
> **Comment:**
> *Question 3: Should $V^1$ be $V^k$ right above equation 9?*
>
> **Response:**
> No, actually the superscript $1$ here was our intended notation to reference that we sample 1 negative edge for every positive edge. The original definition relates back to the paragraph between Eqs.(2) and (3) : $\mathcal{V}_{(i,j)}^{K}$  is the set of negative destination nodes sampled for edge $(v_i,v_j)$, and the size of this set is $K$.
>
> Hence for the scenario of a learnable scaling on the negative graph, with $K=1$, we adopt $\mathcal{V}_{(i,j)}^1$. Hope this helps to clarify, but we can still update the text to make this point more clear if needed.

---

> > ### Comment · Reviewer_32eE · 2023-11-22
> > **Response to authors**
> >
> > Thanks for the detailed clarification, which addressed most of my concerns. The reply to Question 3 is almost unreadable. Can you re-edit it?

---

> ### Author Response · Authors · 2023-11-22
> **Follow-up  Response to Reviewer 32eE**
>
> Thanks for the response.  As for our original reply to Question 3, we have re-edited it.  (There was some strange markdown language issue with this text that does not show up in our local markdown editor; we apologize for the inconvenience and hope this version is sufficiently clear.)

---

### Official Review · Reviewer_TNzh · 2023-10-31

**Soundness:** 2 fair
**Presentation:** 3 good
**Contribution:** 1 poor
**Rating:** 3
**Confidence:** 4

**Summary:**

There are 2 main categories of GNNs for link prediction, node-wise and edge-wise architectures. Node-wise architectures are efficient at test time, model expressiveness is limited such that isomorphic nodes contributing to candidate edges may not be distinguishable. Edge-wise methods form edge-specific subgraph embeddings to enrich representation of pair-wise relationships, disambiguating isomorphic nodes to improve accuracy, but they are computationally expensive. To navigate this trade-off, the authors propose a GNN architecture where nodes are updated following a contrastive and a link prediction loss.

**Strengths:**

-	The authors address an important tradeoff in GNNs for link prediction (predictive power vs computational complexity)
-	The approach has a small time complexity (similar to GCN)

**Weaknesses:**

First of all, thanks for sending your work for review.
I have two general comments that would like you to comment on.

1.	The use of a loss function to generate embeddings that minimize the distance between connected nodes and maximize the distance between disconnected nodes is not new. It’s typically used in the context of self-supervised learning/pre-training, and it has been previously used in the context of link prediction. See:
- You, Y., Chen, T., Sui, Y., Chen, T., Wang, Z., & Shen, Y. (2020). Graph Contrastive Learning with Augmentations. http://arxiv.org/abs/2010.13902
- Qiu, J., Chen, Q., Dong, Y., Zhang, J., Yang, H., Ding, M., Wang, K., & Tang, J. (2020). GCC: Graph Contrastive Coding for Graph Neural Network Pre-Training. Proceedings of the ACM SIGKDD International Conference on Knowledge Discovery and Data Mining, 1150–1160. https://doi.org/10.1145/3394486.3403168
- Shiao, W., Guo, Z., Zhao, T., Papalexakis, E. E., Liu, Y., & Shah, N. (2023). Link Prediction with Non-Contrastive Learning. International Conference on Learning Representations (ICLR). https://openreview.net/pdf?id=9Jaz4APHtWD
- Bordes, A., Usunier, N., Garcia-Durán, A., Weston, J., & Yakhnenko, O. (2013). Translating Embeddings for Modeling Multi-relational Data. Advances in Neural Information Processing Systems (NeurIPS). https://papers.nips.cc/paper_files/paper/2013/file/1cecc7a77928ca8133fa24680a88d2f9-Paper.pdf

My question is, what is the novelty of your approach given that the notion of contrastive learning has previously been used to compute node embeddings?

2.	What you mention as “including negative edges in the forward pass” seems to me as actually just having 2 steps in the backward pass. A first step where node embeddings are updated based on the contrastive loss, and a second step where node embeddings are further updated based on the link prediction loss.

**Questions:**

See weaknesses section

---

> ### Author Response · Authors · 2023-11-19
> **Response to Reviewer TNzh**
>
> **Comment:**
> *Weakness 1: The use of a loss function to generate embeddings that minimize the distance between connected nodes and maximize the distance between disconnected nodes is not new. It’s typically used in the context of self-supervised learning/pre-training, and it has been previously used in the context of link prediction. See [reviewer provided reference list].  My question is, what is the novelty of your approach given that the notion of contrastive learning has previously been used to compute node embeddings?*
>
> **Response:**
> It is definitely true that contrastive learning (and/or self-supervised learning) is commonly employed for producing node embeddings that can be used for link prediction. However, in the references the reviewer listed, as well as in the link prediction literature more broadly, the role of contrastive learning is effectively limited to determining a useful training objective.  But for pipelines that involve a parameterized encoder (i.e., a GNN) for producing embeddings, contrastive learning is independent of the actual encoder architecture itself; rather, it merely defines the loss used to train the encoder.
>
> Turning to our model, YinYanGNN is a specific type of GNN encoder, which can be trained using constrastive pairs for link prediction just as with existing GNN architectures.  However, there remain two fundamental distinctions relative to a typical GNN encoder:
>
> 1. For a standard GNN, the model input is just the original graph and node features, while for YinYanGNN the negative graph serves as an additional model input.  In both cases the actual training loss, which may be contrastive, is the same.
> 2. Unlike a typical GNN, each layer of YinYanGNN computes  node embeddings that, by design, happen to also descend an energy function, namely, Eq.(11).  This aspect introduces a useful inductive bias into the encoder architecture, which is superior to simply inputting the negative graph to a regular GNN, e.g., see our ablation from Section D.3 of the supplementary.
>
> We hope these points help to elucidate the distinctive aspects of YinYanGNN within the landscape of existing methods that use contrastive learning as a training loss.  With respect to the latter, the user-supplied references are indeed relevant alternative approaches for link prediction, even if they do not undercut the novelty of YinYanGNN. As such, during the limited rebuttal window we have quickly conducted an additional experiment with two datasets using the most recent reference [1].  Results are shown in the table below, where we note that, as a self-supervised method, the performance of [1] is significantly lower than YinYanGNN.
>
> | HR@50 |Cora| Citeseer |
> | -------- | -------- | -------- |
> | [1]    |81.6±1.3 | 82.2±1.7|
> | YinYanGNN    |95.7±1.3| 95.5±1.1|
>
> [1] Shiao et al., "Link Prediction with Non-Contrastive Learning International," ICLR 2023.
>
> **Comment:**
> *Weakness 2: What you mention as “including negative edges in the forward pass” seems to me as actually just having 2 steps in the backward pass. A first step where node embeddings are updated based on the contrastive loss, and a second step where node embeddings are further updated based on the link prediction loss.*
>
> **Response:**
> While we are open to alternative interpretations of our approach, we presently cannot see how YinYanGNN can be viewed as reducing to a backward pass with two steps.  Note that in our model, the forward and backward passes serve identical roles as in any other node-wise encoder GNN for link prediction:  In *both* a regular GNN encoder and YinYanGNN, the forward pass computes the node embeddings $Y$, and the backward pass updates only model weights (e.g., $\{W,\theta,\lambda_K \}$ for YinYanGNN), not node embeddings.  The key distinction between them lies with different model inputs and the inductive bias of the respective encoders (see response to first reviewer question above for more details), not the roles of forward and backward passes.
>
>
> Returning to the reviewer's proposed alternative interpretation, the suggestion seems to be that the combined YinYanGNN forward and backward passes can be reduced to a 2-step backward pass such that: Both steps produce new node embeddings, with the second pass refining or updating the embeddings produced by the first pass.  However, unless we have misunderstood this proposal (and please let us know if we have), it seems incompatible with the structure of YinYanGNN as described above.
>
> Regardless, as a side note we would argue that, even if it were possible to characterize YinYanGNN as two integrated backward passes, we do not view this as a bad thing.  Rather, this would just represent another angle for describing and interpreting the model.  Anyway, thanks for raising this topic, as it is interesting to reconsider the foundations of our model.

---

### Official Review · Reviewer_EieC · 2023-11-02

**Soundness:** 3 good
**Presentation:** 2 fair
**Contribution:** 3 good
**Rating:** 5
**Confidence:** 3

**Summary:**

This work studied the research problem of link prediction. The authors discussed the pros and cons of node-wise and link-wise link prediction literature as well as their trade-offs, and then proposed YinYanGNN which has better expressiveness than the existing node-wise methods while still preserving the fast inference speed of node-wise methods.

**Strengths:**

1. The research problem is very interesting given that the link prediction literature has been gradually separated into two groups.
2. The proposed method is reasonably designed and showed promising results across multiple benchmarks, including several larger ones.
3. The authors provided complexity analysis as well as inference time comparison of the proposed method, illustrating the efficiency of the proposed method.

**Weaknesses:**

1. I appreciate the complexity analysis and inference time comparison. Nonetheless, I'd suggest the authors to also include total runtime comparison or even wall times of different procedures for more comprehensive understanding of the efficiency. E.g., Table 3 in [1]
2. Results in Table 2 included strong edge-wise methods with only basic node-wise methods. Even the ones in Table 6 are not specifically designed for link prediction. I'd suggest including more recent / stronger node-wise methods to make the effectiveness comparison more compelling.
3. I found the method section hard to follow even with careful reading

[1] Graph Neural Networks for Link Prediction with Subgraph Sketching, ICLR 2023

**Questions:**

please see above

---

> ### Author Response · Authors · 2023-11-19
> **Response to Reviewer EieC**
>
> **Comment:**
> *Weakness 1: I appreciate the complexity analysis and inference time comparison. Nonetheless, I'd suggest the authors to also include total runtime comparison or even wall times of different procedures for more comprehensive understanding of the efficiency. E.g., Table 3 in [1]*
>
> **Response:**
> Good suggestion.  Using the two largest OGB link prediction graphs, PPA and Citation2, we show the total inference runtime (seconds) in the table below (which we can easily add to the paper if the reviewer prefers).  From these results, we observe that YinYanGNN is considerably faster than edge-wise models, and on par with the node-wise GNNs (GCN and SAGE).
>
>
> | Dataset | SEAL |Neo-GNN |GDGNN |SUREL |SUREL+ |BUDDY |YinYanGNN|GCN| SAGE|
> | -------- | -------- | -------- | -------- | -------- | -------- |-------- | -------- | -------- |-------- |
> | PPA (seconds)    | $\sim 4500$     | 19     |210     |972     |299     |178| 2 | 1.7 |1.7 |
> | Citation2 (seconds)    | $\sim 48000$      | 63     |1680     |11367     |1516     |385|3 | 2.5 |2.4 |
>
>
>
> **Comment:**
> *Weakness 2: Results in Table 2 included strong edge-wise methods with only basic node-wise methods. Even the ones in Table 6 are not specifically designed for link prediction. I'd suggest including more recent / stronger node-wise methods to make the effectiveness comparison more compelling.*
>
> **Response:**
> As it turns out, after the edge-wise SEAL algorithm was originally published, not many node-wise alternatives have been proposed, likely because it has been difficult to achieve superior performance.  For example, there are few node-wise entries atop the OGB link prediction leaderboard; almost all are edge-wise methods or related.
>
> In fact, as of the time of our submission there was no model of any kind with OGB leaderboard performance above ours on all datasets, and *only a single published node-wise model on the leaderboard above ours on any dataset*: this is model CFLP w/ JKNet, and only when applied  to DDI.  But when we examine the CFLP paper, we find that their performance is much lower than YinYanGNN on other non-OGB datasets as shown below:
>
> | HR@20 | Cora | Citeseer |Pubmed |
> | -------- | -------- | -------- | -------- |
> | CFLP w/ JKNet    | 65.57±1.05     | 68.09±1.49     |44.90±2.00    |
> | YinYanGNN    | 84.10±3.23     | 90.52±0.72    |72.38±5.31     |
>
> There is also a single, unpublished node-wise model on the leaderboard called PLNLP worth mentioning.  The PLNLP approach does lead to strong accuracy on both Collab and DDI, but the main contribution is a new training loss that is orthogonal to our method.  In fact any node-wise model, including ours, could be trained with their loss to improve performance.  The only downside is that the PLNLP approach seems to be dataset dependent, with lesser performance on other benchmarks, e.g., PLNLP only achieves 32.38 Hits@100 on PPA.
>
> **Comment:**
> *Weakness 3: I found the method section hard to follow even with careful reading*
>
> **Response:**
> Admittedly, it is challenging to compress sufficient background and supporting analysis within a short conference paper.  As one small improvement, we have updated Algorithm 1 in Figure 1 to include a few more relevant details that link back to the text and may help ground the intended meaning.  We also note that an expended version of Algorithm 1 is included in Section A of the supplementary.

---

### Official Review · Reviewer_pZx7 · 2023-11-08

**Soundness:** 2 fair
**Presentation:** 2 fair
**Contribution:** 2 fair
**Rating:** 5
**Confidence:** 2

**Summary:**

This paper proposes a new link prediction framework that aims at improving the expressive capability of the node-based link predictors. The proposed method, namely YinYanGNN, incorporates an additional graph that is constructed from pure negative edges. While generating node embeddings from both the original graph and the additional graph, YinYanGNN can acquire better expressive capability. The authors connect the reason behind why such an implementation would work from the perspective of energy functions. On top of the energy functions explored by existing works, the energy function explored in this work incorporates an additional term that enforces dissimilarities between neighbors on the negative graph. Experiments over multiple datasets are conducted and the proposed YinYanGNN exhibits promising results.

**Strengths:**

1. I think the overall idea of enhancing node-based link predictors by heuristics from graph-based link predictors is meaningful.

2. The implementation of the proposed  framework seems easy to implement.

**Weaknesses:**

1. The introduction is motivated by improving the expressive capability of node-based link predictors, but the proposed method does not seem to improve that. I think the expressive capability discussed by in frameworks like BUDDY or SEAL is related to isomorphism. And the proposed YinYanGNN does not seem to be related to this concept. Section 5.3 tries to showcase this connection, but I think YinYanGNN only leverages this connection implicitly. I might be understanding this wrong due to the second weakness next.

2. The method section is a little bit difficult to understand. I think the authors should concisely conclude how YinYanGNN works (e.g., how do the forward pass and backward pass look like? what is the model architecture, etc) before introducing the energy function. I feel like the authors explain the overall idea through a series of vague terms (e.g., before section 4.1) that are difficult to understand without sufficient background knowledges.

3. It seems like the training requires learning over multiple negative graphs. This operation seems to introduce a lot of computational overheads during the training.

4. The time complexity analysis seems problematic. A lot of overheads during the training can be moved to the pre-processing.

**Questions:**

Q1: Why does equation (3) need not generally be convex or even lower bounded?

Q2: Do operations described around Equation (13) supports mini-batch training?

---

> ### Author Response · Authors · 2023-11-19
> **Response to Reviewer pZx7 (part 1)**
>
> **Comment:**
> *Weakness 1: The introduction is motivated by improving the expressive capability of node-based link predictors, but the proposed method does not seem to improve that. I think the expressive capability discussed by in frameworks like BUDDY or SEAL is related to isomorphism. And the proposed YinYanGNN does not seem to be related to this concept. Section 5.3 tries to showcase this connection, but I think YinYanGNN only leverages this connection implicitly. I might be understanding this wrong due to the second weakness next.*
>
> **Response:**
> There are many different ways to increase expressiveness and/or break isomorphisms to improve link prediction.  This is especially true for edge-wise methods such as BUDDY or SEAL that benefit from link-specific subgraph information but at a much higher computational cost.  In contrast, for node-wise methods there are more modest options for increasing expressiveness, and our YinYanGNN model uniquely does so by weaving the negative sampling graph into the very fabric of the model architecture (as opposed to merely using negative samples as a training loss).  In this way, although admittedly to a lesser degree than edge-wise methods, some isomorphisms/symmetries can in fact be broken as illustrated in Section 5.3, leading to improved performance as shown in our ablation from Table 3.  Overall then, conditioned on restricting to computationally-efficient node-wise methods, we believe this enhancement is indeed significant, even while differing considerably from prior methods.
>
> **Comment:**
> *Weakness 2: The method section is a little bit difficult to understand. I think the authors should concisely conclude how YinYanGNN works (e.g., how do the forward pass and backward pass look like? what is the model architecture, etc) before introducing the energy function. I feel like the authors explain the overall idea through a series of vague terms (e.g., before section 4.1) that are difficult to understand without sufficient background knowledges.*
>
> **Response:**
> We appreciate the reviewer's suggestion regarding a possible way to reorganize the content to improve accessibility.  However, one issue is that the key starting point of YinYanGNN is actually the energy function from Eq.(2) which is later refined to Eq.(11).  If we were to instead present the model architecture itself first, where each YinYanGNN model layer is given by Eq.(13), the structure seems very ad hoc.
>
> Alternatively, one potentially useful upgrade is to refine Algorithm 1 and the attendent text from Section 4.3 to provide more details and clarify.  In summary here though, once we accept Eq.(13) as the description of the YinYanGNN architecture, where $t$ is the layer index, then the forward pass is simply executing (13) for $T$ steps/layers.  The backward pass is exactly like any other multi-layer GNN, only here the parameter set is merely $\{ W, \theta \}$ and optionally $\lambda_K$ ; please see (13) for the role of these parameters within the YinYanGNN layers.
>
>
> **Comment:**
> *Weakness 3: It seems like the training requires learning over multiple negative graphs. This operation seems to introduce a lot of computational overheads during the training.*
>
> **Response:**
> During training, we sample $K$ negative graphs for each epoch; however, the time spent on sampling is inconsequential relative to the overall training time per epoch. Hence the $K$-dependent complexity, as noted in Table 1 for YinYanGNN, stems from the forward and backward pass computations, where $K$ has the effect of increasing the number of edges proportionally.  For smaller graphs this is still computationally cheap, but for larger graphs we fix $K=1$ while learning the coefficient $\lambda_K$.  As shown in the ablation from Section D.1 of the supplementary, this allows us to roughly match the performance of larger $K$ models while only using a more efficient $K=1$ model.

---

> ### Author Response · Authors · 2023-11-19
> **Response to Reviewer pZx7 (part 2)**
>
> **Comment:**
> *Weakness 4: The time complexity analysis seems problematic. A lot of overheads during the training can be moved to the pre-processing.*
>
> **Response:**
> For SEAL and BUDDY, the training complexity is taken directly from reference (Chamberlain et al., 2023) and appears to be correct while accounting for possible pre-processing.  As for YinYanGNN, we break down the training complexity $ O(T|\mathcal{E}|Kd+nP d^2+|\mathcal{E}_{tr}|d^2)$ as follows. First, $T|\mathcal{E}|Kd+nP d^2$ is the time required to compute a single forward pass to produce embeddings for every node.  We remark that the $K$ factor in this expression is because YinYanGNN uniquely uses the negative graph as input, and is equivalent to making the original graph larger; it is not because of sampling time or any other factor that could be moved to pre-processing.
>
> And secondly, $ |\mathcal{E}_{tr}|d^2 $ is the time complexity of the HadamardMLP decoder used by YinYanGNN for computing the loss as needed by the backward pass. Note that the backward pass has equivalent complexity to the forward pass, and hence has already been accounted for in the above expressions.
>
> **Comment:**
> *Question 1: Why does equation (3) need not generally be convex or even lower bounded?*
>
> **Response:**
> Eq.(3) has three quadratic terms, the first two are convex and lower-bounded.  In contrast, the last term is concave an unbounded from below.  To see this, note that $L^-$ is a PSD matrix and hence the factor $tr[Y^\top L^- y]$ must be non-negative and convex.  As such, because there is a minus sign in front of this term, it flips from convex to concave, and as $Y$ becomes large the term can be driven to minus infinity.  Thus the overall function need not be convex nor bounded from below, depending on the values of $\lambda$ and $\lambda_K/K$, which balance the relative contribution of each term.  However, if we explicitly bound the last term of Eq.(3) from below using an operator like softplus, as we do in Eqs.(6) and (11), then we can at least ensure that the energy will always be bounded from below (although it still need not always be convex).
>
> **Comment:**
> *Question 2: Do operations described around Equation (13) supports mini-batch training?*
>
> **Response:**
> Great question, we should have mentioned this.  But the short answer is yes: YinYanGNN directly supports mini-batch training.  In this way it is just like any other node-wise model, and can incorporate the training enhancements or scalability heuristics as applied to standard architectures such as GCN.